# Equivariant neural networks and equivarification

### Abstract

Equivariant neural networks are special types of neural networks that preserve some symmetries on the data set. In this paper, we provide a method to modify a neural network to an equivariant one, which we call *equivarification*.

## 1 Introduction

One key issue in deep neural network training is the difficulty of tuning parameters, especially when the network size grows larger and larger Han et al. (2015). In order to reduce the complexity of the network, many techniques have been proposed by analyzing the structural characteristics of data, for example, sparsity Wen et al. (2016), invariance in movement Goodfellow et al. (2009).

One of the most important structural characteristics of data is symmetry. By utilizing the translation symmetry of an object in the photo, convolutional neural network (CNN)(Krizhevsky et al. (2012)) uses shared filters to reduce the number of parameters compared with fully connected networks. However, to handle the case of rotation and reflection, people usually use the data augmentation approach to generate additional input data that has a bunch of training images with different rotation angles of the original images.

In contrast to data augmentation approach, another idea is to design more sophisticated neural networks, such that the input data with certain symmetries can be trained together and applied to reduce the training complexity. Recent attempts have been made in the equivariant CNN Cohen & Welling (2016a); Cohen et al. (2018); Weiler et al. (2018). These existing works target at special cases and cannot be easily generalized to arbitrary networks and arbitrary symmetries.

In this paper, we take advantage of the symmetry of the data set and provide a general method to modify an arbitrary neural network so that it preserves the symmetry. The process is called *equivarification*. A key feature is that our equivarification method can be applied without detailed knowledge of a layer in a neural network, and hence, can be generalized to any feedforward neural networks. Another feature is that the number of parameters in the new neural network that we need to train is the same as the original one, if we equivarify the original network globally (see the second paragraph in Section 4 for details). In addition, we can also output how each data instance is changed from the canonical form (for example, in an image classification problem, additionally we can also output how many degrees an image is rotated from original upside-up image) using the same network.

We rigorously prove that our equivarification method produces truely equivariant neural networks. Practically, we equivarify a CNN as an illustration. We conduct experiments using the MNIST data set where the resulting equivariant neural network predicts the number and also the angle. If we forget the angle and keep only the number, we get an invariant neural network.

### 1.1 Motivation

By the symmetry of the data set, we mean a group action (see Definition 2.2) on the data set. Let us consider a simple cat image classification example. Then the data set can be the set of all images of a fixed size. Rotation symmetry means that we can rotate an image and the resulting image still lies in the data set. In other words, the group of rotations acts on the data set.

One can build a cat classifier that assigns a number between 0 and 1 to each image indicating the probability that it is an image of a cat. If one image is a rotation (say by 90 degree counterclockwise)

of another one, then it makes sense to require a classifier to assign the same probability to these two images. A classifier that satisfies this property is said to be *invariant* under 90-degree rotation, which is a special case of being *equivariant* under 90-degree rotation. To give an example of an (non-invariant) equivariant neural network, we furthermore require our classifier not only produces the probability of being a cat image, but also outputs an angle, say in $\{0, 90, 180, 270\}$ (more precisely, the probability of each angle). Suppose that the equivariant classifier predicts that an image is rotated by 90 degrees, then it should predict a 270-degree rotation for the same image but rotated by 180 degrees.

Not only does it make sense to require the cat classifier to be equivariant, but also it is more "economical" to be equivariant if implemented correctly. Roughly speaking, a regular classifier "treats" an image and its rotated ones as separated things, but ideally an equivariant neural network "sees" their connections and "treats" them together. From another point of view, given a regular neural network, if we apply the data augmentation method carefully, since the training data is symmetric, it is possible that after training, the network is (approximately) equivariant (depending on the initialization). The fact that it is equivariant implies that there is now symmetry among the parameters. For example, some parameters are the same as other parameters, so there is redundancy. While in our equivariant neural network, the equivariance of our neural network is built into the structure of the neural network by sharing parameters, and in particular, it is independent of the loss function, initialization, and the data set. For instance, for the training data, it does not make any difference in results whether we prepare it by randomly rotating and recording the angles, or not rotating and labeling everything as degree 0.

In our approach, to make a neural network equivariant, at each layer other than the input layer we add a bunch of neurons (multiplying by the order of the group), but we don't introduce new parameters. Instead the added neurons share parameters with the original ones.

## 1.2 RELATED WORK

Invariance in neural networks has attracted attention for decades, aiming at designing neural networks that capture the invariant features of data, for example, face detection system at any degree of rotation in the image plane Rowley et al. (1998), invariance for pattern recognition Barnard & Casasent (1991), translation, rotation, and scale invariance Perantonis & Lisboa (1992).

Recently, several works have started to look into equivariant CNN, by studying the exact mapping functions of each layer Cohen & Welling (2016a;b); Cohen et al. (2018); Marcos et al. (2017); Lenssen et al. (2018); Cohen et al. (2019), and some symmetries are studied such as translation symmetry, rotation symmetry. These methods have been used in different application domains, such as in remote sensing Marcos et al. (2018), digital pathology Veeling et al. (2018), galaxy morphology prediction Dieleman et al. (2015).

There are also works that aiming to construct an equivariant neural network without having to specify the symmetry Sabour et al. (2017).

As far as we know, our construction provides the first truly equivariant neural network (See Section 5.1).

## 2 PRELIMINARIES

In this section, we talk about some basics in group theory, like group actions, equivariance, etc. For those who would like to get a more close look at the group theory and various mathematical tools behind it, please refer to any abstract algebra books, for example, Lang (2002).

Here, we first give a couple of definitions about groups in order to help readers quickly grasp the concepts.

**Definition 2.1.** A group $(G, \cdot)$ consists of a set $G$ together with a binary operation "·" (which we usually call multiplication without causing confusing with the traditional sense of multiplication for real numbers) that needs to satisfy the four following axioms.

1) Closure: for all $a, b \in G$, the multiplication $a \cdot b \in G$.

2) Associativity: for all $a, b, c \in G$, the multiplication satisfies $(a \cdot b) \cdot c = a \cdot (b \cdot c)$.

3) Identity element: there exists a unique identity element $e \in G$, such that, for every element $a \in G$, we have $e \cdot a = a \cdot e = a$.

4) Inverse element: for each element $a \in G$, there exists an element $b \in G$, denoted $a^{-1}$, such that $a \cdot b = b \cdot a = e$, where $e$ is the identity element.

Note that in general, commutativity does not apply here, namely, for $a, b \in G$, $a \cdot b = b \cdot a$ does not always hold true.

For example, all integers with the operation addition $(\mathbb{Z}, +)$. One can easily check that the four axioms are satisfied and $0$ is the identity element.

As another example, a group consists of a set $\{0, 1\}$ together with the operation $+$ (mod 2) where $0+1 = 1+0 = 1$ and $1+1 = 0+0 = 0$. The identity element is $0$. When applying this to the image processing tasks, this can be interpreted as follows. $1$ represents the action of rotating an image by $180^o$ and $0$ represents the action of not rotating an image. Then $0 + 1$ represents the combination of operations that we first rotate an image by $180^o$ and then keep it as it is, so that the final effect is to rotate an image by $180^o$; while $1 + 1$ represents that we first rotate an image by $180^o$ and then rotate again by $180^o$, which is equivalent to that we do not rotate the original image.

Similarly, we can define the element $1$ as the operation of flipping an image vertically or horizontally, which we can give similar explanations for the group.

Therefore, instead of using translations, rotations, flippings, etc., we can use abstract groups to represent operations on images, and hence, we are able to design corresponding equivariant neural networks disregarding the operations of images (symmetries of data) and just following the group representation.

In the following, we give a definition about group actions. Let $X$ be a set, and $G$ be a group.

**Definition 2.2.** We define a $G$-action on $X$ to be a map

$$T : G \times X \to X$$

(on the left) such that for any $x \in X$

- $T(e, x) = x$, where $e \in G$ is the identity element,

- for any $g_1, g_2 \in G$, we have

$$T(g_1, T(g_2, x)) = T(g_1 g_2, x).$$

Frequently, when there is no confusion, instead of saying that $T$ is a $G$-action on $X$ or equivalently that $G$ acts on $X$ via $T$, we simply say that $G$ acts on $X$; and $T$ is also understood from the notation, i.e., instead of $T(g, x)$ we simply write $gx$, and the above formula becomes $g_1(g_2 x) = (g_1 g_2)x$.

We say $G$ acts trivially on $X$ if $gx = x$ for all $g \in G$ and $x \in X$.

Let $X, Y$ be two sets, and $G$ be a group that acts on both $X$ and $Y$.

**Definition 2.3.** A map $F : X \to Y$ is said to be $G$-equivariant, if $F(gx) = gF(x)$ for all $x \in X$ and $g \in G$. Moreover, if $G$ acts trivially on $Y$ then we say $F$ is $G$-invariant.

*Example* 2.4. Let $X$ be the space of all images of $28 \times 28$ pixels, which contains the MNIST data set. Let $G$ be the cyclic group of order $4$. Pick a generator $g$ of $G$, and we define the action of $g$ on $X$ by setting $gx$ to be the image obtained from rotating $x$ counterclockwise by 90 degrees. Let $Y$ be the set $\{0, 1, 2, ..., 9\} \times \{0, 90, 180, 270\}$. For any $y = (\text{num}, \theta) \in Y$ we define

$$gy := (\text{num}, (\theta + 90)\text{mod}_{360}).$$

An equivariant neural network that classifies the number and rotation angle can be viewed as a map $F$ from $X$ to $Y$. Equivariance means if $F(x) = (\text{num}, \theta)$ then $F(gx) = (\text{num}, (\theta + 90)\text{mod}_{360})$, for all $x \in X$.

Thus, we can see that we can model each layer in the neural network as a group $G$ that acts on a set $X$, where we can interpret the set $X$ as input to this layer and the group action as the function mapping or operation of this layer. By abstracting the behaviors on the original input data using groups

(for example, using a same group to represent either rotation by $180^o$ and flipping, or even more abstract operations during intermediate layers), we are able to apply group actions on different sets $X_1, X_2, X_3, \ldots$ (where each one represents input to different layers) and design similar equivariant network structures based on an original one.

## 3 EQUIVARIFICATION

In this section, we talk about the detailed method of performing equivarification and its theoretical foundation. This is the key part of the paper to understand our proposed equivarification method. Those who would like to avoid mathematical proofs can directly go to the examples we provide to get intuitive ideas of how we construct the equivariant neural networks.

In this section we fix $X$ and $Z$ to be two sets, and $G$ to be a group that acts on $X$.

**Definition 3.1.** Define the $G$-product of $Z$ to be

$$Z^{\times G} = \{s : G \to Z\},$$

the set of all maps from $G$ to $Z$.

We define a $G$-action on $Z^{\times G}$ by

$$G \times Z^{\times G} \to Z^{\times G}$$
$$(g, s) \mapsto gs,$$

where $gs$ as a map from $G$ to $Z$ is defined as

$$(gs)(g') := s(g^{-1}g'), \tag{3.1}$$

for any $g' \in G$.

We have the projection map $p : Z^{\times G} \to Z$ defined by

$$p(s) = s(e), \tag{3.2}$$

for any $s \in Z^{\times G}$ where $e \in G$ is the identity element. Then

**Lemma 3.2.** *For any map $F : X \to Z$, there exists a unique $G$-equivariant map $\widehat{F} : X \to Z^{\times G}$ such that $p(\widehat{F}(x)) = F(x)$ for all $x \in X$.*

*Proof.* For any $x \in X$, we define $\widehat{F}(x)$ as a map from $G$ to $Z$ by

$$(\widehat{F}(x))(g) = F(g^{-1}x),$$

for any $g \in G$. To see that $\widehat{F}$ is $G$-equivariant, we need to check for any $x \in X$ and $g \in G$, $\widehat{F}(gx) = g(\widehat{F}(x))$. For any $h \in G$, $(\widehat{F}(gx))(h) = F(h^{-1}gx)$ by the definition of $\widehat{F}$, while $(g(\widehat{F}(x)))(h) = (\widehat{F}(x))(g^{-1}h) = F(h^{-1}gx)$. We leave the proof of uniqueness to the readers. $\quad\square$

*Remark* 3.3. In the Definition 3.1 and Lemma 3.2, $G$ is an arbitrary group and $Z$ is an arbitrary set. It is easy to see that we can adjust them, if we consider other categories. For example, when $G$ is a compact Lie group and $Z$ is a differentiable manifold, we can re-define $Z^{\times G}$ to be the space of differentiable maps from $G$ to $Z$; when $G$ is a non-compact Lie group and $Z$ is a differentiable manifold, we can consider the space of compact supported smooth maps. When we implemented the neural network for infinite $G$, we have to approximate it by a finite subset of $G$. Then in this case, we need to work in the realm of approximately equivariant. For this implementation reason, we restrict our group $G$ to be a finite group for the rest of the paper.

This lemma can be summarized as the commutative diagram in Figure 1. It motivates the following general definition.

**Definition 3.4.** We say a tuple $(\widehat{Z}, T, p)$ a $G$-equivarification of $Z$ if

- $\widehat{Z}$ is a set with a $G$-action $T$;

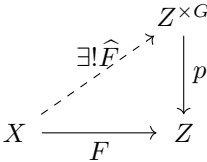

Figure 1: For any $X$ and $F$, there exists a unique lift $\widehat{F}$ such that the diagram commutes.

- $p$ is a map from $\widehat{Z}$ to $Z$;
- For any set $X$ with a $G$-action, and map $F : X \to Z$, there exists a $G$-equivariant map $\widehat{F} : X \to \widehat{Z}$ such that $p \circ \widehat{F} = F$.

Here $\circ$ denotes the composition of maps. As usual, $T$ will be omitted from notation.

In Section 4 we will see applications of $G$-equivarification to neural networks. From Lemma 3.2 we know that the triple of the $G$-product $Z^{\times G}$, the $G$-action defined in the Formula 3.1, and the projection $p$ defined in Formula 3.2 is a $G$-equivarification. There are other $G$-equivarifications. See Appendix for more discussion.

*Example* 3.5. Let $G$ be the cyclic group of order 4. More concretely, we order elements of $G$ by $(e, g, g^2, g^3)$. The set $Z^{\times G}$ can be identified with $Z^{\times 4} = Z \times Z \times Z \times Z$ via the map

$$s \mapsto (s(e), s(g), s(g^2), s(g^3)). \tag{3.3}$$

Then $G$ acts on $Z^{\times 4}$ by $g(z_0, z_1, z_2, z_3) = (z_3, z_0, z_1, z_2)$, and the projection map $p : Z^{\times 4} \to Z$ is given by $(z_0, z_1, z_2, z_3) \mapsto z_0$. Let $F : X \to Z$ be an arbitrary map, then after the identification $\widehat{F}$ becomes a map from $X$ to $Z^{\times 4}$ and

$$\widehat{F}(x) = (F(x), F(g^{-1}x), F(g^{-2}x), F(g^{-3}x)).$$

One can check that $\widehat{F}$ is $G$-equivariant. The map $p$ is given by

$$p(z_0, z_1, z_2, z_3) = z_0.$$

It is easy to see that $p \circ \widehat{F} = F$.

## 4 APPLICATION TO NEURAL NETWORKS

In this section, we show through an example how our proposed equivarification method works.

Let $\{L_i : X_i \to X_{i+1}\}_{i=0}^n$ be an $n$-layer neural network (which can be CNN, multi-layer perceptrons, etc.). In particular, $X_0$ is the input data set, and $X_{n+1}$ is the output data set. Let $G$ be a finite group that acts on $X_0$. Let $L$ be the composition of all layers

$$L = L_n \circ L_{n-1} \circ \cdots \circ L_0 : X_0 \to X_{n+1}.$$

Then we can equivarify $L$ and get maps $\widehat{L} : X_0 \to \widehat{X}_n$ and $p : \widehat{X}_{n+1} \to X_{n+1}$. Then $\widehat{L}$ is an equivariant neural network.

Alternatively, one can construct an equivariant neural network layer by layer. More precisely, the equivariant neural network is given by $\{\widehat{L_i \circ p_i} : \widehat{X}_i \to \widehat{X}_{i+1}\}_{i=0}^n$, where $\widehat{L_i \circ p_i}$ is the equivarification of $L_i \circ p_i$ for $i \in \{0, 1, ..., n\}$, $\widehat{X}_0 = X_0$ and $p_0 = \mathrm{id}$ is the identity map (See Figure 2). More precisely, by the commutativity of Figure 2 we know that

$$p_{n+1} \circ \widehat{L_n \circ p_n} \circ \widehat{L_{n-1} \circ p_{n-1}} \circ \cdots \circ \widehat{L_0 \circ p_0} = L = p \circ \widehat{L}.$$

Then both $\widehat{L_n \circ p_n} \circ \widehat{L_{n-1} \circ p_{n-1}} \circ \cdots \circ \widehat{L_0 \circ p_0}$ and $\widehat{L}$ are equivarifications of $L$. Suppose that for both equivarifications we have chosen $\widehat{X}_{n+1}$ to be $X_{n+1}^{\times G}$. Then by the statement in Theorem 3.2, we have

$$\widehat{L_n \circ p_n} \circ \widehat{L_{n-1} \circ p_{n-1}} \circ \cdots \circ \widehat{L_0 \circ p_0} = \widehat{L}.$$

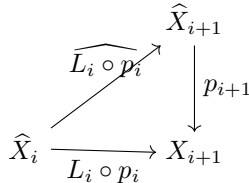

Figure 2: Equivarification layer by layer.

Sometimes, other than equivarifying the map $L_i \circ p_i : \widehat{X}_i \to X_{i+1}$, it makes sense to construct some other map $L_i'$ from $\widehat{X}_i$ to some other set $X_{i+1}'$, and then we can equivarify $L_i'$. This makes the equivariant neural network more interesting (see the example below).

*Example* 4.1. Let the 0-th layer $L_0 : X_0 \to X_1$ of a neural network that is defined on the MNIST data set be a convolutional layer, and $X_1 = \mathbb{R}^{\ell_1}$, where $\ell_1 = 28 \times 28 \times c_1$, and $c_1$ is the number of channels (strides = $(1,1)$, padding = 'same'). Let $G = \{e, g, g^2, g^3\}$ be the cyclic group of order 4 such that $g$ acts on $X_0$ as the 90-degree counterclockwise rotation. Then we construct $\widehat{L}_0 : X_0 \to \mathbb{R}^{4\ell_1}$ by

$$x_0 \mapsto (L_0(x_0), L_0(g^{-1}x_0), L_0(g^{-2}x_0), L_0(g^{-3}x_0)).$$

For the next layer, instead of equivarifying $L_1 \circ p_1 : \mathbb{R}^{4\ell_1} \to \mathbb{R}^{\ell_2}$, we can construct another convolutional layer directly from $\mathbb{R}^{4\ell_1}$ by concatenating the four copies of $\mathbb{R}^{\ell_1}$ along the channel axis to obtain $\mathbb{R}^{28 \times 28 \times 4c_1}$, and build a standard convolution layer on it. This new construction of course changes the number of variables compared to that of the original network.

From the above analysis and Lemma 3.2, it is not hard to derive the following summary.

**Main result** *Let $\mathbf{X} = \{L_i : X_i \to X_{i+1}\}_{0 \le i \le n+1}$ be an original neural network that can process input data $\{x_0^j\}_j \subseteq X_0$ and labelling data $\{x_{n+1}^j\}_j \subseteq X_{n+1}$. Let $G$ be a finite group that acts on $X_0$. The proposed $G$-equivarification method is able to generate a $G$-equivariant neural network $\widehat{\mathbf{X}} = \{\widehat{L}_i : \widehat{X}_i \to \widehat{X}_{i+1}\}_{0 \le i \le n+1}$ that can process input data $\{x_0^j\}_j \subseteq X_0 = \widehat{X}_0$ and enhanced labeling data $\{\widehat{x}_{n+1}^i\}_j \subseteq \widehat{X}_{n+1}$. Furthermore, the number of parameters of $\widehat{\mathbf{X}}$ is the same as that of $\mathbf{X}$.*

## 5 EXPERIMENTS

In this section, we show our experiments on the MNIST dataset, which achieve promising performance[1]

Our designed equivariant neural network using the proposed equivarification method is shown in Figure 3. Note that equivarification process does not increase the number of variables. In our case, in order to illustrate flexibility we choose not to simply equivarify the original neural network, so the layer conv2 and conv3 have four times the number of variables compared to the corresponding original layers.

Next, we discuss the labeling of the input data. Since the neural network is $G$-equivariant, it makes sense to encode the labels $G$-equivariantly. Suppose $(x_0, x_{n+1}) \in X_0 \times X_{n+1}$ is one labelled data point. Then in the ideal case, one hopes to achieve $L(x_0) = x_{n+1}$. Assuming this, to give a new label for the data point $x_0$ for our equivariant neural network we need to define $\widehat{x}_{n+1} = \widehat{L}(x_0)$. For this, it is sufficient to define $\widehat{L}(x_0)(g)$ for all $g \in G$. By equivariance, $\widehat{L}(x_0)(g) = L(g^{-1}x_0)$. If $g = e$ then $\widehat{L}(x_0)(g) = L(x_0) = x_{n+1}$. If $g \ne e$, it is very likely that the data $g^{-1}x_0$ originally does not have a label, so we do not know ideally what $L(g^{-1}x_0)$ should be. In the naive data augmentation approach, $L(g^{-1}x_0)$ is labeled the same as $L(x_0)$ hoping to get $L$ as close to an invariant map as possible. In our case, we do not have such restriction, since we do not need $L$ to be invariant. In

---

[1]The code in tensorflow is uploaded as the supplementary material. In the code, we also include a version that allows 90 degree rotation and horizontal and vertical flips, just to make the group non-commutative.

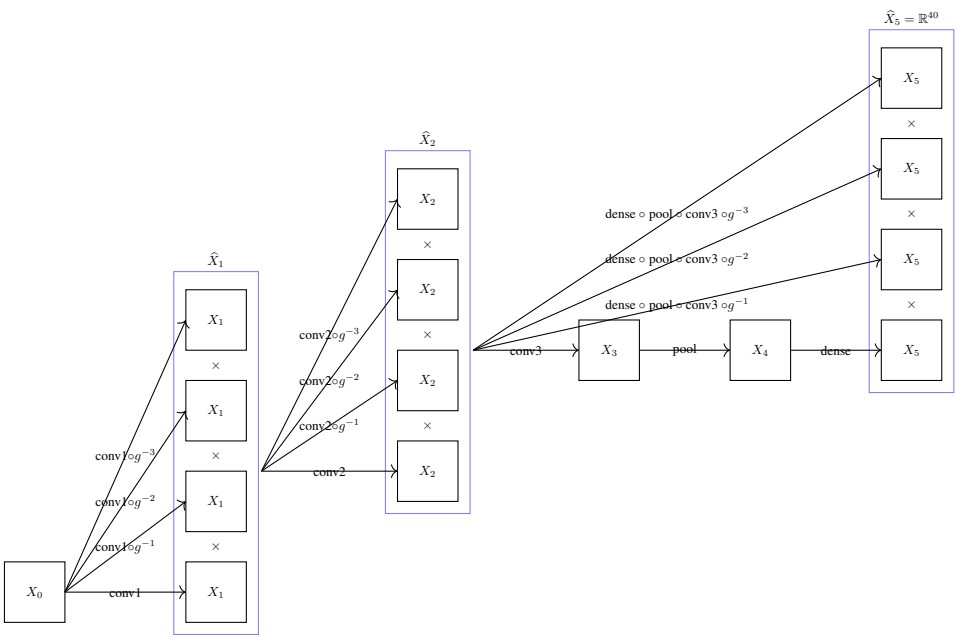

Figure 3: In this figure, conv1 is a standard convolution layer with input = $X_0$ and output = $X_1$. After equivarification of conv1, we get four copies of $X_1$. Then we stack the four copies along the channel direction, and take this whole thing as an input of a standard convolution layer conv2. We equivarify conv2, stack the four copies of $X_2$, and feed it to another convolution layer conv3. Now instead of equivarifying conv3, we add layer pool and layer dense (logistic layer), and then we equivarify their composition dense $\circ$ pool $\circ$ conv3 $\circ g^{-1}$ and get $\widehat{X}^5 = \mathbb{R}^{40}$. To get the predicted classes, we can take an argmax afterwards.

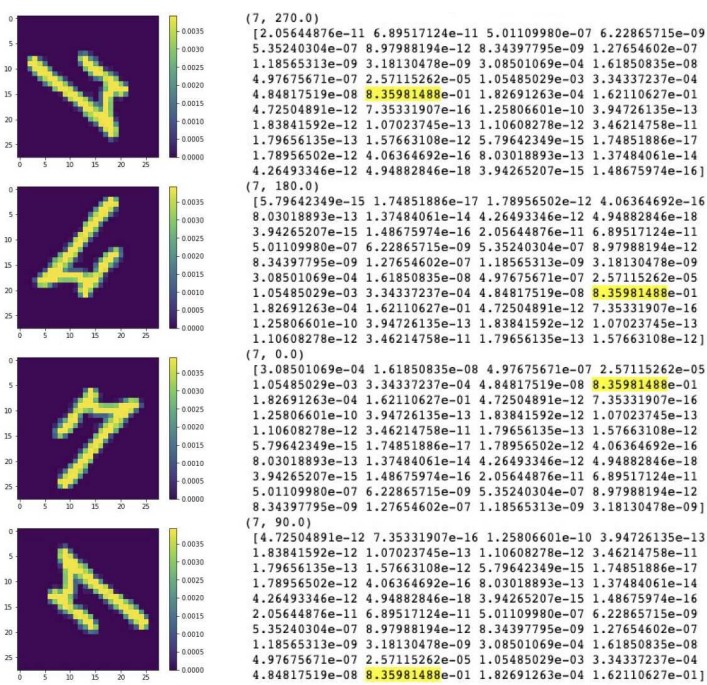

Figure 4: On the left, we have the rotated images; on the right, we have the predicted numbers, angles, and the probability vectors in $\mathbb{R}^{40}$, each component of which corresponds to the probability of a (number, angle) combination. The equivariance in this case means that the four vectors are related by shifts.

practice, $X_{n+1}$ is a vector space, and we choose to label $L(g^{-1}x_0)$ by the origin of $X_{n+1}$. In our MNIST example, this choice is the same as the following.

For $m \in \{0, 1, 2, ..., 9\}$ denote

$$e_m = (0, \cdots, 0, \underset{\underset{m\text{-th spot}}{\uparrow}}{1}, 0, \cdots, 0) \in \mathbb{R}^{10}.$$

For an unrotated image $x_0 \in X_0$ that represents the number $m$, we assign the label $e_m \oplus 0 \oplus 0 \oplus 0 \in \mathbb{R}^{40}$. Then based on the equivariance, we assign

$$\begin{aligned} gx_0 &\mapsto 0 \oplus e_m \oplus\ 0\ \oplus\ 0, \\ g^2 x_0 &\mapsto 0 \oplus\ 0\ \oplus e_m \oplus\ 0, \\ g^3 x_0 &\mapsto 0 \oplus\ 0\ \oplus\ 0\ \oplus e_m. \end{aligned}$$

For each testing image in the MNIST data set, we randomly rotate it by an angle of degree in $\{0, 90, 180, 270\}$, and we prepare the label as above. For the training images, we can do the same, but just for convenience, we actually do not rotate them, since it won't affect the training result at all.

## 5.1 EQUIVARIANCE VERIFICATION

To spot check the equivariance after implementation, we print out probability vectors in $\mathbb{R}^{40}$ of an image of the number 7 and its rotations. We see that the probability vectors are identical after a shift by 10 slots. See Figure 4.

## 5.2 ACCURACY

Here we count the prediction as correct if both the number and the angle are predicted correctly. The accuracy of our neural network on the test data is $96.8\%$. This is promising when considering the fact that some numbers are quite hard to determine its angles, such as $0, 1$, and $8$.

## 6 CONCLUSION

In this paper, we proposed an equivarification method to design equivariant neural networks that are able to efficiently process input data with symmetries. Our proposed method can be generalized to arbitrary networks or functions by leveraging group action, which enables our design to be uniform across layers of feedforward neural networks, such as multi-layer perceptrons, CNNs, without being aware of the knowledge of detailed functions of a layer in a neural network. As an illustration example, we show how to equivarifying a CNN for image classification. Results show that our proposed method performs as expected, yet with significantly reduction in the design and training complexity.

## A  APPENDIX - MORE ABOUT EQUIVARIFICATION

In Section 3 we define $(Z^{\times G}, p)$ as an example of $G$-equivarification. In this section, we show that it is "minimal" in the sense of its universal property.

**Lemma A.1** (universal property). *For any $G$-equivarification $(\widehat{Z}', p')$ of $Z$, there exists a $G$-equivariant map $\pi : \widehat{Z}' \to Z^{\times G}$ such that $p' = p \circ \pi$. Moreover, for any set $X$ and map $F : X \to Z$, the lifts $\widehat{F} : X \to Z^{\times G}$ and $\widehat{F}' : Z \to \widehat{Z}'$ of $F$ satisfy $\pi \circ \widehat{F}' = \widehat{F}$. (See Figure 5.)*

*Proof.* We define the map $\pi : \widehat{Z}' \to Z^{\times G}$ by $[\pi(\widehat{z}')](g) = p'(g^{-1}\widehat{z}')$, where $\widehat{z}' \in \widehat{Z}'$ and $g \in G$. To show $p' = p \circ \pi$, for any $\widehat{z}' \in \widehat{Z}$, we check $p \circ \pi(\widehat{z}') = p[\pi(\widehat{z}')] = [\pi(\widehat{z}')](e) = p'(\widehat{z}')$. To show $\pi$ is $G$-equivariant, for any $\widehat{z}' \in \widehat{Z}$, and $h \in G$, we compare $\pi(h\widehat{z}')$ and $h\pi(\widehat{z}')$: for any $g \in G$, $[\pi(h\widehat{z}')](g) = p'(g^{-1}h\widehat{z}')$ and $[h\pi(\widehat{z}')](g) = [\pi(\widehat{z}')](h^{-1}g) = p'(g^{-1}h\widehat{z}')$. Lastly, we show $\pi \circ \widehat{F}' = \widehat{F}$. Note that $\pi \circ \widehat{F}'$ is a $G$-equivariant map from $X$ to $Z^{\times G}$, and

$$p \circ (\pi \circ \widehat{F}') = p' \circ \widehat{F}' = F,$$

so by the uniqueness of Lemma 3.2, we get $\pi \circ \widehat{F}' = \widehat{F}$.  □

Now we discuss about finding a "smaller" equivarification in another direction, shrinking the group by bring in the information about $X$. Let $N = \{g \in G \mid gx = x \text{ for all } x \in G\}$, the set of elements in $G$ that acts trivially on $X$. It is easy to check that $N$ is a normal subgroup of $G$. We say $G$ acts on $X$ effectively if $N = \{e\}$. In the case when $G$ does not act effectively, it makes sense to consider the $G/N$-product of $Z$, where $G/N$ is the quotient group. More precisely, consider $Z^{\times G/N} = \{s : G/N \to Z\}$, which is smaller in size than $Z^{\times G}$. For any map $F : X \to Z$, we can get a $G/N$-equivariant lift $\widehat{F}$ of $F$ following the same construction as Lemma 3.2 (with $G$ replaced by $G/N$). Since $G$ maps to the quotient $G/N$, we have that $G$ acts on $Z^{\times G/N}$ and $\widehat{F}$ is also $G$-equivariant.

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

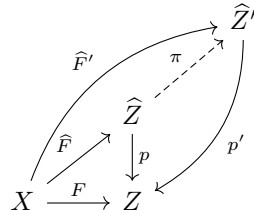

Figure 5: Factors through.

Taco S Cohen, Maurice Weiler, Berkay Kicanaoglu, and Max Welling. Gauge equivariant convolutional networks and the icosahedral cnn. *arXiv preprint arXiv:1902.04615*, 2019.

Sander Dieleman, Kyle W Willett, and Joni Dambre. Rotation-invariant convolutional neural networks for galaxy morphology prediction. *Monthly notices of the royal astronomical society*, 450 (2):1441–1459, 2015.

Ian Goodfellow, Honglak Lee, Quoc V Le, Andrew Saxe, and Andrew Y Ng. Measuring invariances in deep networks. In *Advances in neural information processing systems*, pp. 646–654, 2009.

Song Han, Huizi Mao, and William J Dally. Deep compression: Compressing deep neural networks with pruning, trained quantization and huffman coding. *arXiv preprint arXiv:1510.00149*, 2015.

Alex Krizhevsky, Ilya Sutskever, and Geoffrey E Hinton. Imagenet classification with deep convolutional neural networks. In *Advances in neural information processing systems*, pp. 1097–1105, 2012.

Serge Lang. *Algebra, GTM211*. Springer, 2002.

Jan Eric Lenssen, Matthias Fey, and Pascal Libuschewski. Group equivariant capsule networks. In *Advances in Neural Information Processing Systems*, pp. 8844–8853, 2018.

Diego Marcos, Michele Volpi, Nikos Komodakis, and Devis Tuia. Rotation equivariant vector field networks. In *Proceedings of the IEEE International Conference on Computer Vision*, pp. 5048–5057, 2017.

Diego Marcos, Michele Volpi, Benjamin Kellenberger, and Devis Tuia. Land cover mapping at very high resolution with rotation equivariant cnns: Towards small yet accurate models. *ISPRS journal of photogrammetry and remote sensing*, 145:96–107, 2018.

Stavros J Perantonis and Paulo JG Lisboa. Translation, rotation, and scale invariant pattern recognition by high-order neural networks and moment classifiers. *IEEE Transactions on Neural Networks*, 3(2):241–251, 1992.

Henry A Rowley, Shumeet Baluja, and Takeo Kanade. Rotation invariant neural network-based face detection. In *Proceedings. 1998 IEEE Computer Society Conference on Computer Vision and Pattern Recognition (Cat. No. 98CB36231)*, pp. 38–44. IEEE, 1998.

Sara Sabour, Nicholas Frosst, and Geoffrey E Hinton. Dynamic routing between capsules. In I. Guyon, U. V. Luxburg, S. Bengio, H. Wallach, R. Fergus, S. Vishwanathan, and R. Garnett (eds.), *Advances in Neural Information Processing Systems 30*, pp. 3856–3866. Curran Associates, Inc., 2017. URL http://papers.nips.cc/paper/6975-dynamic-routing-between-capsules.pdf.

Bastiaan S Veeling, Jasper Linmans, Jim Winkens, Taco Cohen, and Max Welling. Rotation equivariant cnns for digital pathology. In *International Conference on Medical image computing and computer-assisted intervention*, pp. 210–218. Springer, 2018.

Maurice Weiler, Fred A Hamprecht, and Martin Storath. Learning steerable filters for rotation equivariant cnns. In *Proceedings of the IEEE Conference on Computer Vision and Pattern Recognition*, pp. 849–858, 2018.

Wei Wen, Chunpeng Wu, Yandan Wang, Yiran Chen, and Hai Li. Learning structured sparsity in deep neural networks. In *Advances in neural information processing systems*, pp. 2074–2082, 2016.

