# OpenReview forum: "Equivariant neural networks and equivarification"
_ICLR.cc/2020/Conference — Reject_

### Official Review · AnonReviewer3 · 2019-10-18
**Official Blind Review #3**

**Rating:** 6

**Review:**

In this work, the authors employ concepts from group theory to turn an arbitrary feed forward neural network into an equivariant one, i.e. a network whose output transforms in a way that is consistent with the transformation of the input. To this end, the authors first introduce the basic concepts of group theory required to follow their work and provide a comprehensive definition of equivariance. They then explain how to equivarify (w.r.t. a finite group G) a given neural network, and present experimental results on rotated MNIST digits to support their approach.

I find that the proposed approach is very elegant and addresses a highly important question, namely how to devise or modify an architecture such that it preserves symmetries. In my opinion, the current paper makes an interesting theoretical contribution towards achieving this goal, but it also has several shortcomings that are detailed below. Based on these shortcomings, I recommend rejecting the paper but I would be willing to increase the score if these points were addressed in sufficient detail.

Major comments:

1) Scaling
The authors mention in the abstract that ‘although the network size scales up, the constructed equivariant neural network does not increase the complexity of the network compared with the original one in terms of the number of the parameters.’ Based on Eq. (3.0.2) and Fig. 3, it is my understanding that n evaluations of each data point (input to a layer) are required for a cyclic group of order n. If the outputs of a layer are then concatenated, the input dimension of the subsequent layer grows by a factor of n. I would therefore argue that out-the-box application of the proposed approach does increase the number of variables dramatically and that the abstract is misleading in this respect. The authors briefly comment on this point with one sentence in the third paragraph of the introduction. However, I would appreciate if this point was addressed in more detail, for example in a dedicated paragraph after the theory is introduced. Please also address the question of whether variable sharing is essential from an equivariance point of view, or whether it’s simply a necessity to prevent an explosion of the number of parameters. Furthermore, convolutions encode translational symmetry which may be beneficial for the current application but may not be desirable for other datasets. A few comments for clarification would be very helpful. Finally, the equivarified network seems to increase the required number of computations significantly compared to the original one, which I find worth a comment .

2) Experiment
The authors only consider a convolutional architecture and say that ‘in order to illustrate flexibility we choose not to simply equivarify the original network’. However, to me one of the main advantages of the paper seems to be that you can take this approach to equivarify any FFN. It would therefore be interesting to see this approach be applied to different networks starting with a simpler one, e.g. a 2-layer MLP. The authors could then compare the original network to the equivarified one with and without variable sharing. That would not only help the reader understand the approach better but also be much more in line with the main motivation of the paper. Then adding a second experiment, e.g. a convolutional architecture, to demonstrate flexibility would be very interesting. With regard to Fig. 4, I think there may be better ways of summarising the results than dumping 160 numbers of which only 4 seem to be of interest. The message seems to be that the network yields identical probabilities irrespective of the degree of rotation. What I find surprising is that all numbers are actually identical (shifted by 10). Is this by construction?

3) Limitations
As indicated in the second paragraph of Sec. 4, this approach is limited to finite groups and the authors only consider image rotations w.r.t. the cyclic group of degree 4. Although I appreciate that this is meant to serve as a toy problem to illustrate that the approach works, I do not think that rotations by a constant angle are very interesting. What would be really interesting is equivariance w.r.t continuous rotations (Lie Groups), e.g. the SO(2) in this particular case. I doubt that an extension to the SO(2) is straightforward within the current theoretical framework. However, even if that is the case, I would appreciate if the authors could comment on this in a paragraph.

Minor comments:

i) There are many typos and grammar mistakes in the paper:
‘any feedforward networks’ -> ‘any feedforward network’.
‘enables to design’ -> ‘enables us to design’
‘our proposed equivarification method use’ -> ‘our proposed equivarification method uses’
‘traditional sense multiplication’ -> ‘traditional sense of multiplication’
‘a group G acts’ -> ‘a group G that acts’
‘neural network that defined on the’ -> ‘neural network that is defined on the’
‘which achieves promising performance’ -> ‘which achieve promising performance’
‘supplymentary material’ -> ‘supplementary material’
Etc.

ii) I think there may be a mistake in the 3rd point of Definition 3.0.3: For consistency with the previous definitions and with Fig. 1, shouldn’t F map from X to Z and \hat F from X to \hat Z?

iii) Last paragraph of Sec 3: ‘then after the identification \hat F becomes a map from Z to...'.  Should it be ‘a map from X to ..’?

iv) In Definition 3.0.3 you define the tuple (\hat Z, T, p) to be a G-equivarification, but in the paragraph below you call the G-product itself a G-equivarification (without including T and p).

v) Footnote 2: You could correct for that and present the theory shifting by g instead of g^-1 to make it easier for the reader to follow. Or, at least, give a reference to the footnote earlier on in Example 4.0.1 to avoid confusion.

vi) Unless there is a special reason for this, I would suggest changing the enumeration of definitions, lemmas, examples and equations, i.e. (3.0.1) -> (3.1), etc...


*********************************************************
I increased my rating based on the authors addressing many of my comments.
*********************************************************


**Experience Assessment:**

I do not know much about this area.

**Review Assessment: Checking Correctness Of Derivations And Theory:**

I assessed the sensibility of the derivations and theory.

**Review Assessment: Checking Correctness Of Experiments:**

I assessed the sensibility of the experiments.

**Review Assessment: Thoroughness In Paper Reading:**

I read the paper thoroughly.

---

> ### Author Response · Authors · 2019-11-06
> **respond to Official Blind Review #3**
>
> We really appreciate your time reading the paper and the constructive comments.
>
> 1) Scaling
>
> A. You are right. If only do a global equivarification using a group of order n, then the number of parameters is the same as the original one. However, if we concatenate, then the number of variables grow by a factor of n. We will address this more precisely in the revised version (coming soon).
>
> We were not trying to mislead readers (even though it is a little confusing.). We want to mention that to simply achieve equivariant, we can do a global equivarification of the neural network (not a layer by layer one), and this naive version does not increase number of parameters, but as you mentioned it does seem increase the computation complexity.
>
>
> 2) Experiment
>
> A. The experiments you suggested make sense. However, since we are planning to equivarify a few well-known neural networks, such as vgg, resnet in a following up paper (with additional authors) and do more comparison there, we do not include many experiments in this paper. I don't think there is surficent amount of time for us in this tight time period to do a clear and fair comparison to include into this paper.
>
> With regard to Fig. 4, I agree with what you said that it is not the best way to present data. But we do have our own reasons: I have read a few other articles about equivariant neural networks, and to be frank, I don't fully understand how they achieve the equivariance. I am including this picture also as a suggestion to test equivariance in neural networks. One does not need to read the paper or check the code. One can simply print out the all the probability predictions for one image and its rotations, then we can check whether it is equivariant or not. Yes, they are all shifted by 10 locations by construction. In the code, https://github.com/symplecticgeometry/equivariant-neural-networks-and-equivarification/blob/master/src/equivariant_cnn%20V2.ipynb
> we included the horizontal and vertical flips and one can see that the probabilities are also shifted, but in a more complicated way.
>
>
> 3) Limitations
>
> A. That is a good point. We will add a comment about it. In the paper, the example is only based on cylic group of order 4. In the code, we also have the dehedral4 group (rotation by 90, horizontal and vertical flips) of order 8. Even for finite groups, there are already a lot of applications. In NLP, for certain languagues, there are masculine vs. feminine symmetries (a group of order 2). Since this equivarification method is not picky about original neural networks, it does seem to have a lot of applications.
>
> As for other groups, Lemma 3.0.2 does not require any thing about finiteness. For a topological group, it is natural to require that the Z^{\times G} in the definition 3.0.1 to be the space of continuous maps from G to Z; for a Lie group, we can use the space of smooth maps from G to Z; for non-compact Lie groups, we can use the space of compactly supported smooth maps from G to Z. It is the implementation part that is less trivial. For implementation purpose, all the continuous groups need to be approximated by a finite thing (maybe a subgroup). We are fine with the approximation, but of course, after this approximation, we should only hope approximately equivariance instead of strictly equivariance. We did not set up the math framework to allow approximately equivariant. (A type of neural network is approximately equivariant, if as we approximate the group finer and finer, it becomes more and more equivariant.) All we wanted for this paper is to set up some simple foundations for equivariant neural network, and then people build more sophiscated things on top of it. Also for non-compact Lie groups, the approximate equivariant neural network is more subtle.
>
>
> Minor comments:
>
> i)
> A. Thanks a lot! We will change accordingly.
>
> ii)
> A. Sorry for the terrible typo.
>
> iii)
> A. Sorry for the terrible typo.
>
> iv)
> A. Will change accordingly.
>
> v)
> A. In the new code in github, we correct it to agree with the paper.
>
> vi)
> A. Will change accordingly.

---

### Official Review · AnonReviewer1 · 2019-10-23
**Official Blind Review #1**

**Rating:** 3

**Review:**

In this paper, the authors propose a method for making a neural network equivariant. Their method also can be applied to make each layer equivariant too.

Strengths:
-- The paper is very well written and easy to follow with clear notation.

-- The derivations seem to be correct.


Weaknesses:
-- The experiment is nice but very limited and does not demonstrate the benefits of having an equivariant network. For example, the authors do not report the accuracy of recovering the original (0) rotation.

-- The novelty of the work is questionable. While the development is different, the final example for equivarification of a neural network is very similar to the existing works by Cohen and Welling.

-- There are other works on equivarification that are missed by this paper. For example, consider the following paper:
Lenssen, J. E., Fey, M., & Libuschewski, P. (2018). Group equivariant capsule networks. In NeurIPS.

-- The layer-wise equivariant method does have extra computational overheads.

-- The fact that we have to specify the groups that we want to make the network equivariant with respect to is a limitation. The promise of capsule networks, in contrast, is to "ideally" learn the pose (variation) vectors in a data-driven way.
Sabour, S., Frosst, N., & Hinton, G. E. (2017). Dynamic routing between capsules. In NeurIPS.

-- The following statements need more explanation:
  * "However, these may require extra training overhead as the augmented data increase."

**Experience Assessment:**

I have read many papers in this area.

**Review Assessment: Checking Correctness Of Derivations And Theory:**

I did not assess the derivations or theory.

**Review Assessment: Checking Correctness Of Experiments:**

I assessed the sensibility of the experiments.

**Review Assessment: Thoroughness In Paper Reading:**

I read the paper at least twice and used my best judgement in assessing the paper.

---

> ### Author Response · Authors · 2019-11-06
> **Respond to Official Blind Review #1**
>
> We appreciate the time you spent reviewing this paper. We are sorry that some unfortunate typos make the paper harder to read.
>
> Weaknesses:
> -- The experiment is nice but very limited and does not demonstrate the benefits of having an equivariant network. For example, the authors do not report the accuracy of recovering the original (0) rotation.
>
> A. We are planning to do a careful analysis of the experiments and comparisons in a following up paper.
>
> -- The novelty of the work is questionable. While the development is different, the final example for equivarification of a neural network is very similar to the existing works by Cohen and Welling.
>
> A. We have read a few existing works by Cohen and Welling, and we also cited their papers. Actually it is their spherical cnn paper that brought us into this area. But frankly speaking, we don't understand how they achieve equivariance. The only point in their papers that we understand is to achieve invariance by taking average (in the spherical cnn paper, taking average means integrating against a Haar measure). But even this, it is not so clear to me how it is carriered out. For this reason, we could not justify the novelty of our paper since I don't understand what is in their papers. On the other hand, we propose a criterion to test equivariance without understanding a paper or reading a code. In Figure 4 of this paper, we print out all the predicted probabilities. And one can see that all the four 40-dimensional vectors are exactly related by shifting. We marked out one component by yellow, but it is true for all other components. Our neural network certainly passed this test. One can view this from this code on github
> https://github.com/symplecticgeometry/equivariant-neural-networks-and-equivarification/blob/master/src/equivariant_cnn%20V2.ipynb
> There, we also include horizontal flip and vertical flip, and one gets a vector in R^80. One can check that for the same image under different rotations/flips, the probabilities are the same except the shifts.
>
> With these being said, our best understanding of the difference of our paper compared to the Cohen and welling's papers is:
> they try to achieve invariance by taking average; while, we achieve equivariance by enlarging the space.
>
> -- There are other works on equivarification that are missed by this paper. For example, consider the following paper:
> Lenssen, J. E., Fey, M., & Libuschewski, P. (2018). Group equivariant capsule networks. In NeurIPS.
>
> A. Thanks a lot bring up this paper. We will cite it properly in the revision (coming in 1-2 days.)
>
> -- The layer-wise equivariant method does have extra computational overheads.
> A. Agree.
>
> -- The fact that we have to specify the groups that we want to make the network equivariant with respect to is a limitation. The promise of capsule networks, in contrast, is to "ideally" learn the pose (variation) vectors in a data-driven way.
> Sabour, S., Frosst, N., & Hinton, G. E. (2017). Dynamic routing between capsules. In NeurIPS.
>
> A. That is quite interesting. We will cite it.
>
> -- The following statements need more explanation:
>   * "However, these may require extra training overhead as the augmented data increase."
> A. We will make it more clear. We think what we wrote here is confusing. In a task that just to predict the numbers not the angles, our neural network (the plain version) has the approximately computing complexity as the data augmentation one. But because we guarantee invariance (a special type of equivariance), while data augmentation does not, we are using the neurons more "economically".

---

### Official Review · AnonReviewer4 · 2019-11-04
**Official Blind Review #4**

**Rating:** 3

**Review:**

Motivated by group action theory, this paper proposes a method to obtain ‘equivariant’ neural nets given trained one, where ‘equivariant’ refers to a network that gives identical output if certain symmetry of the dataset is performed on the input (for example, if we rotate a sample the predicted class should not change).

After reading the paper, I don’t understand the experiments section. In particular, it is not clear to me how the proposed method differs from regular data augmentation, as to my understanding, the input to conv1 is copied 4 times and performed rotation for 0, 90, 180 and 270 degrees and the 4 times increased number of parameters (in-depth) of conv1 are shared. Furthermore, the same rotations are performed to the input to the second layer-conv 2: as the augmentation is cyclic I don’t understand why the authors perform this operation second time. Could the authors elaborate on this? After reading this section, I don’t understand the proposed fine-tuning procedure (pre-train, finetune and test): (1) what is the accuracy of the pre-trained network that was started from? (2) how is the initial network fine-tuned and modified? (as the authors mention that during training the samples are not rotated). Also, I am confused with the first sentence on page 8: ‘the complexity of the constructed network does not grow in terms of the number of parameters’. It would be useful if the results in Fig. 4 are more clearly illustrated.

Is the order or increased computation 4x4x4? It would be useful to compare the method (computation & performance) with a baseline where the dataset is enlarged with data augmentation. The authors mention in the introduction that this increases training overhead, whereas the proposed practical method increases the computation at inference as well as the memory footprint of the model and the forward pass. It would be useful if the authors compared empirically with baselines with (1) data augmentation (2) network with an increased number of parameters (same as the proposed one).

In summary, the idea of using group action theory seems interesting. However after reading the paper, it is not clear to me how the idea is carried out, and although the authors provide theoretical justification, it is not clear how this connects with the practical proposed method and whether it outperforms standard data augmentation (see above). Moreover, I find the writing of the paper quite unclear (see discussion above and examples below).

- If digits 6 or 9 are rotated the label changes, how does the proposed method handle this?
- page 8, conclusion: The authors claim that the proposed approach yields a ‘significant reduction in the design and training complexity’. I don’t understand relative to what this comparison refers to, as the regular data augmentation approach is more straightforward in my opinion. Also, given that this is pointed as an important contribution, in my opinion, an empirical comparison must be done with such baselines (see above).
- Page 1: it is mentioned that ‘the number of parameters can be the same as the original network’ but the experiments do not include such architecture. After reading the paper I don’t understand how such a network can be implemented and whether it works.

— Minor —
- Page 1 & 1par-Pg2: I don’t understand what the authors mean by ‘uniformly *across layers* of NN?
- Page 2: In these existing works, … I don’t understand this sentence
- Page 2: our .. method use -> uses
- Page 2: map over the orbits. I don’t understand this
- Page 2: the first truly equivariant NN. After reading Sec 5 I don’t understand this point.
- Sec. 4: how to equivarifying -> equivarify
- Page 4: ‘pick a generator’, would recommend elaborating this term or only mentioning g as an element of G for clarity for readers unfamiliar with group theory
- What is the testing accuracy if rotated for different angles than trained (e.g. 45 degrees)?

**Experience Assessment:**

I have published one or two papers in this area.

**Review Assessment: Checking Correctness Of Derivations And Theory:**

I assessed the sensibility of the derivations and theory.

**Review Assessment: Checking Correctness Of Experiments:**

I carefully checked the experiments.

**Review Assessment: Thoroughness In Paper Reading:**

I read the paper thoroughly.

---

> ### Author Response · Authors · 2019-11-07
> **respond to Official Blind Review #4**
>
> Thank you very much for your time reviewing this paper. We really appreciate all the comments. We are sorry that there are some typos in the current version which makes the reading unnecessarily harder. We are fixing this in the next version (coming in 1-2 days) and we are also making changes reflecting your valuable comments.
>
> Q. After reading the paper, I don’t understand the ... Could the authors elaborate on this?
>
> 	A. this method vs data augmentation:
> 	Data augmentation does not guarantee equivariance.
>
> 	In the plain version of the equivarification method, we don't need to do it layer by layer as you suggested.
>
> 	As for the experiment that we carried out, in short, to guarantee equivariance, we make each layer equivariant. More details, for the plain version, doing a layer by layer equivarification is equivalent to do a global equivarification.
>
> Q. After reading this section, I don’t understand the proposed fine-tuning procedure (pre-train, finetune and test): (1) what is the accuracy of the pre-trained network that was started from?
>
> 	A. By pre-train accuracy, do you mean the accuracy on the training data before fine tune? Since this is a baby model, we actually did not check that. We only look at the loss in the training case. We also did not do the fine tune. Since our main goal is not to provide a specific equivariant neural network for image recognization, but to provide a general process to produce equivariant neural networks by modifying existing networks. Also, we are planing to do a full analysis and comparison in a following up paper.
>
> Q. (2) how is the initial network fine-tuned and modified? (as the authors mention that during training the samples are not rotated). Also, I am confused with the first sentence on page 8: ‘the complexity of the constructed network does not grow in terms of the number of parameters’. It would be useful if the results in Fig. 4 are more clearly illustrated.
>
> Is the order or increased computation 4x4x4? It would be useful to compare the method (computation & performance) with a baseline where the dataset is enlarged with data augmentation. The authors mention in the introduction that this increases training overhead, whereas the proposed practical method increases the computation at inference as well as the memory footprint of the model and the forward pass. It would be useful if the authors compared empirically with baselines with (1) data augmentation (2) network with an increased number of parameters (same as the proposed one).
>
> 	A. The initial network is not fine tuned. We just want to provide a general method to modify neural networks and we also did not carry out careful and fair comparison of our method against others (such as data augmentation). We will make the sentence ‘the complexity of the constructed network does not grow in terms of the number of parameters’ clearer in the revised version. We will make the Fig. 4 more clearly illustrated.
>
> - If digits 6 or 9 are rotated the label changes, how does the proposed method handle this?
>
> 	A. we don't label the rotated 6 the same as an unrotated 9.
>
> - page 8, conclusion: The authors claim ... with such baselines (see above).
>
> 	A. agree. We will change the wording. The paper is mainly a proposed method with a theoretical proof and an experiment justification. It does not deal with the comparison with previous work, which requires a lot of experiments and tests, otherwise, finding one or two examples and comparing the results won't be fair.
>
> - Page 1: it is mentioned that ‘the number of parameters can be the same as the original network’ but the experiments do not include such architecture. After reading the paper I don’t understand how such a network can be implemented and whether it works.
>
> 	A. The experiment include a more interesting example, in which case, it is hard to compare the original network and the new network.
>
> 	On the other hand, one can apply lemma 3.0.3 to the whole neural network, i.e., make F to be the original neural network. Then in the proof of lemma 3.0.3, the construction of \hat F is our equivarification. Clearly \hat F has the same number of parameters as F. For detailed tensorflow implementation, see https://github.com/symplecticgeometry/equivariant-neural-networks-and-equivarification/blob/master/src/equivariant_cnn%20V2.ipynb
>
> 	[106] drop X1_hat, X2_hat, and rewrite X6_hat as
>
>     X6_hat = equivarification(X0_hat, graphs, 'basic_graph',
>                               ['conv3', 'dense', 'logi'],
>                               layer_paremeters,
>                               'permutation')
>
> — Minor —
> A. Will change these accordingly.

---

### Official Review · AnonReviewer2 · 2019-11-08
**Official Blind Review #2**

**Rating:** 3

**Review:**

The paper adds an interesting new perspective to equivariant neural nets. However, the actual construction looks equivalent to steerable neural nets to me (see the papers by Cohen and Welling). The generalization of steerable nets has been published under the name "gauge equivariant neural nets", it would be very interesting to chart out the exact connections between these concepts.

The authors mention that Z^{\times G} is not the only possible lifting space. I believe that the general case would be Z^V where V is a representation of G.

Many of the earlier papers on equivariant nets were written in the language of representation theory. It is interesting that similar nets can be constructed by purely group theoretic methods, but I really think that ultimately they are same thing. Consequently, I would expect all the experimental results to be identical.

What would make this paper really valuable for didactic purposes is if these connections were carefully mapped out and presented with intuitive diagrams and examples.

**Experience Assessment:**

I have published in this field for several years.

**Review Assessment: Checking Correctness Of Derivations And Theory:**

I carefully checked the derivations and theory.

**Review Assessment: Checking Correctness Of Experiments:**

I assessed the sensibility of the experiments.

**Review Assessment: Thoroughness In Paper Reading:**

I read the paper thoroughly.

---

> ### Author Response · Authors · 2019-11-09
> **respond to Official Blind Review #2**
>
> Thanks for the feedback. I just looked at the two papers you mentioned (steerable one and the gauge one). Unfortunately, all the papers look very similar in this area. It is hard to tell exact what is going on in a paper without a rigorous definition. (of course, I will cite them).
>
> The main contribution of our paper is bringing in the  Definition (3.0.1 or 3.1 in the coming version) of a G-product, and Lemma 3.0.2 (or 3.2 in the coming version).
>
> When you say Z^V, I do not understand what exactly you mean. Do you mean V is a vector space, and there is a group homomorphism (a representation) form G to the GL(V), the space of general linear transforms on V. If so, what is Z^V?
>
> In our paper, Z^{\times G} is the space of maps from G to Z. In the case when G is a compact Lie Group, we can restrict  Z^{\times G} to be the space of smooth maps from Z to G to make it smaller and respect the differential structures. When G is a non-compact, we can further restrict to the compact supported maps.
>
> It would be ideal that we can read all papers in this field and make it clear what the connections are. But on the other hand, this paper is not doing a special case study, and we have to worry about whether our paper is a special case of other papers. We provide a general and simple framework for equivariant network. I briefly go through all the papers that are mentioned to me, and I don't see any construction as general as this one.
>
> I want to stress that, the definition 3.1 and lemma 3.2 are the cleanest thing that I have seen in this equivariant area. We did not make any additional and unnecessary choices.

---

### Public Comment · ~K_V_Subrahmanyam1 · 2019-11-07
**Recent References to be included.**

The authors should also consult the recent work of Kondor (Risi Kondor. N-body networks: a covariant hierarchical neural network architecture for learning atomic potentials)  and Murugan et al (SO(2)-equivariance in Neural networks using tensor nonlinearity, BMVC 2019) where other ideas for group equivarification are considered.

---

> ### Author Response · Authors · 2019-11-09
> **respond to references**
>
> I am reading the papers you mentioned. But I only see the only pdfs. Do you have the bibtex? Thanks.

---

### Author Response · Authors · 2019-11-10
**about revision and general respond to all the reviews**

We made a revision. This is not the final version. We are willing to make further modifications if necessary to make the paper more readable, and more fair to other works.

About the changes we made:
    We corrected some terrible typos in the theory part and added some remarks to make it more readable. Most of the changes are marked in the blue color.

    We largely rewrote the introduction part. In particular, we removed those sentences that are confusing. We added more
    heuristic arguments to compare our approach again the data augmentation approach.

    We cited more works, and tried to make it more fair.

    We added an acknowledgement.

Our general respond the referees' comments:

Thanks for all your time and patience. We really appreciate. We find the comments are fair and honest. We find our paper is much user friendly after revision based on your comments.

     1. difference between our approach vs data augmentation:
          They are definitely quite different. In short, using data augmentation, the network is in general not equivariant. We
          added more arguments in the revised version.

    2. experiment limitation:
          The experiment that we carry out is mainly to justify visually the equivariance of our network. It is a toy model. We
          believe that there will be a lot of cool applications of our theory in the future. After all, we feel that the main
          contribution of the paper is theoretical.

    3. novelty:
        Theoretically, I think the approach is the most general one.
        In many other papers, when they say equivariant, they actually mean invariant. But we are not.

        F is a map from X to Z, where X has a group G action.
            To achieve invariance, one can take an average (or other aggregation) of F over the group action.
            To achieve equivariance, we choose to enlarge the space Z.

   4. sanity:
        The theory part is short and easy to check. The experiment outputs vectors in Fig 4 where all vectors are related by
        shifts. In the github code jupyter notebook v2, one can also see the version with flips added in, where the 80
       dimensional vectors are also related by shifts.

    5. limitation on the group:
        The theory works for all groups, but the implementation is only for finite groups (see newly added Remark 3.3).

---

### Decision · Program_Chairs · 2019-12-19

**Decision:**

Reject

**Comment:**

This paper proposes a way to construct group equivariant neural networks from pre-trained non-equivariant networks. The equivarification is done with respect to known finite groups, and  can be done globally or layer-wise. The authors discuss their approach in the context of the image data domain. The paper is theoretically sound and proposes a novel perspective on equivarification, however, the reviewers agree that the experimental section should be strengthened and connections with other approaches (e.g. the work by Cohen and Welling) should be made clearer. The reviewers also had concerns about the computational cost of the equivarification method proposed in this paper. While the authors’ revision addressed some of the reviewers’ concerns, it was not enough to accept the paper this time round. Hence, unfortunately I recommend a rejection.